# Decadal Continuous Meteor-Radar Estimation of the Mesopause Gravity Wave Momentum Fluxes over Mohe: Capability Evaluation and Interannual Variation

Xu Zhou [1,2], Xinan Yue [1,2,3,*], Libo Liu [1,3,4], You Yu [1,2,3], Feng Ding [1,2,3], Zhipeng Ren [1,2,3], Yuyan Jin [1,2,3] and Hanlin Yin [1,2,3]

1   Key Laboratory of Earth and Planetary Physics, Institute of Geology and Geophysics, Chinese Academy of Sciences, Beijing 100029, China
2   Beijing National Observatory of Space Environment, Institute of Geology and Geophysics, Chinese Academy of Sciences, Beijing 100029, China
3   College of Earth and Planetary Sciences, University of Chinese Academy of Sciences, Beijing 100049, China
4   Heilongjiang Mohe National Geophysical Observatory, Institute of Geology and Geophysics, Chinese Academy of Sciences, Beijing 100029, China
*   Correspondence: yuexinan@mail.iggcas.ac.cn

**Abstract:** In the present work, the momentum fluxes of gravity wave (GW) around the mesopause are estimated, using the decadal continuous observations by meteor radar at Mohe ($53.5°$N, $122.3°$E). Applying the Hocking's (2005) approach with the modified-composite-day (MCD) analysis, the GW momentum fluxes of short-periods (less than 2 h) are estimated month by month. As the first step, several experiments are designed to evaluate the accuracy and uncertainty in the estimation. The results show that Mohe meteor radar has the ability to give reasonable estimations on the GW momentum fluxes at a height of 82–94 km, in which errors are generally less than 5 $m^2/s^2$. The uncertainty induced by different angular information of the detected meteor in each month achieves ~2 $m^2/s^2$. It is inferred that the variability of the GW momentum fluxes over 2 $m^2/s^2$ can be distinguished in the observation. The interannual variation of the estimated GW momentum fluxes show a significant enhancement in 2012, and a depression in 2013, with a fluctuation over $\pm10$ $m^2/s^2$ at 82 km. However, no obvious quasi-biennial oscillation (QBO) -like signal has been found in the Lomb–Scargle periodogram.

**Keywords:** gravity wave; momentum fluxes; meteor radar





## 1. Introduction

Atmospheric gravity waves (GWs) are those small-scale disturbances with a large frequency, which play an important role on the atmospheric circulation. As vertically propagating from the lower atmosphere, GWs dissipate in the mesosphere and lower thermosphere and deposit momentum into mean flows. The GW momentum deposition generally reverses the background zonal wind around the mesopause, deviating from the radiative equilibrium [1]. Deepening the understanding of the variability of the GW momentum fluxes with time scales varying from days to years, is persistently a hot topic. Previous observational works have revealed that the GW variability should be modulated by the atmospheric variabilities from below, such as the diurnal atmospheric tides [2], the seasonal variation of background winds [3,4], the Madden–Julian Oscillation (MJO) [5], and the quasi-biennial oscillation (QBO) [6,7]. The proper parameterization of the GW horizontal momentum fluxes ($\overline{u'w'}$, $\overline{v'w'}$) is also necessary for the global circulation models (GCMs), because most of the small-scale waves are sub-grid and thus unresolved [8]. Pedatella et al. [9] suggested that the parameterized GW momentum forcing should be the major source of the bias and uncertainty between the different whole atmosphere models. Thus,

estimating the GW momentum fluxes accurately and investigating its variability not only gives a better understanding of physics but also gives impetus to the model development.

Various measurement techniques have the capability to estimate the GW momentum fluxes, including satellites [10,11], incoherent scatter radars [12,13], airglow images [14], lidars [15,16], and meteor radars [17–19]. Satellite has advantages in the spatial coverage. For example, Ern et al. [20], for the first time, provided the global map of the GW momentum fluxes, based on CRISTA satellite observations of temperature, according to the polarization relations. Ground-based Lidar observation also enable to detect the temperature perturbations [21,22] and have the potential to derive the GW momentum fluxes using the Ern et al. [20] method. Meteor radars, the direct measurement of radial wind velocity, provide continuous observations at a given location, so they are suitable candidates for long-term monitoring on the GW momentum fluxes, which are distinguished from other measurements that have advantages, in case studies. Hocking [17] firstly proposed a widely accepted approach that deals with the radial wind velocity, to estimate the GW momentum flux by the all-sky meteor radars. Hocking's approach is a generalization of the dual-beam method used by Vincent and Reid [23]. The accuracy of the GW estimation is dependent on the meteor counts detected by the meteor radar [19]. Following Hocking's approach, the estimations of the GW momentum fluxes were performed by meteor-radar stations worldwide, and revealed various GW variabilities over different locations [2,4,18,24–26]. For example, Andrioli et al. [27], de Wit et al. [3], and Jia et al. [28] studied the different features of the seasonal variation of the GW momentum flux, based on multi-year meteor-radar observations in Brazil, Norway, and China, respectively. In particular, Fritts et al. [18] examined the capability of Southern Argentina's Agile Meteor Radar (SAAMER) in estimating the gravity wave momentum fluxes. Different from the standard meteor radar, the SAAMER has multiple transmit beams focusing the energy close to the zenith direction, so that it can ensure more reliable measurements. Generally, the ability to observe robust and resilient vertical velocity fluctuations is key and is one of the disadvantages when applying Hocking's method to the standard meteor radars. Using the meteor-radar estimated GW information, together with a gravity-wave model, Pramitha et al. [29] further traced back the GW sources. These investigations improved our understanding of the GW variabilities and their origins. The Mohe (53.5°N, 122.3°E) meteor-radar station operated by the Institute of Geology and Geophysics, Chinese Academy of Sciences (IGGCAS), started observations in August 2011 and uninterruptedly operated for over 10 years with high quality data. The dataset accumulated during the past decade provides a valuable opportunity to quantitatively evaluate the accuracy and uncertainty of the GW momentum fluxes and investigate the interannual variability of the GW momentum fluxes.

As previously discussed, the meteor-radar estimations of the GW momentum fluxes around the mesopause, improve the knowledge of atmospheric physics. However, the accuracy and uncertainty of the measurements rely on the meteor-radar configuration and the detected meteor information, and different meter radars should have a distinct performance. Thus, it is necessary to evaluate the capability of the GW estimation by each specific meteor radar before studying the GW variability. In this paper, we will evaluate the capability of the Mohe meteor radar in estimating the GW momentum fluxes during the last decade firstly, based on a modeling experiment similar to Fritts et al. [18]. Our work will give a comprehensive analysis on how well the Mohe meteor radar on the estimation of the GW momentum fluxes. Then, the estimated GW momentum fluxes in the last decade and their interannual variability will be further discussed. To our knowledge, rare studies give continuous estimation of the GW momentum fluxes over ten years so far. The paper will be organized as follows: Section 2 will give an overview of the Mohe meteor radar and the methodology of the data processing. Section 3 will analyze the capability of estimating the GW momentum fluxes by the Mohe meteor radar in detail and then present the results of the GW variability. Sections 4 and 5 will give a discussion and summary, respectively.

## 2. Materials and Methods

Mohe (53.5°N, 122.3°E) is a mid-latitude location between the tropics and the Arctic polar vortex. The Mohe meteor detection radar has been continuously operated since August 2011. The radar is a conventional atmospheric radar systems (ATRAD) with the transmitted frequency of 38.9 MHz and the peak power of 20 kW. In this study, we select the data within the zenith angle of 15°–60° and with a radial velocity less than 200 m/s. The data with a small zenith angle would induce large error in the estimation of the horizontal wind, and those with a large zenith angle should lead to large uncertainties in the height determination. The selection criterion of the zenith angle is slightly different in previous works, and here we follow the works by Jia et al. [28] and Tian et al. [22,30]. The selected data is distributed in the heights of 70–110 km with a peak around 91 km (Figure 1a). The useful data amounts in the 1-km bin around 90 km for each month, achieves over 50,000 in the summer and guarantees ~20,000 in winter. The meteor radar enables operation for both daytime and nighttime, continuously, and generally undisrupted by the severe weather conditions. The meteor counts reach a maximum at dawn (~03 LT) and about one half around dusk (~18LT). The minimum of the data count in a 1-hr bin exceeds 5000 for every month, during the ten years (Figure 1b). The clear seasonal and diurnal variation of the meteor counts is a general consensus, associating with the natural change in the meteor income rate. The azimuth-zenithal distribution shows that the data count increases with the zenith angle and the quasi-isotropic in the azimuth with slightly more meteors from the north-west direction (Figure 1c). The condition in each month should adequately satisfy the requirement on estimating the GW momentum fluxes, as Vincent et al. [19] proposed.

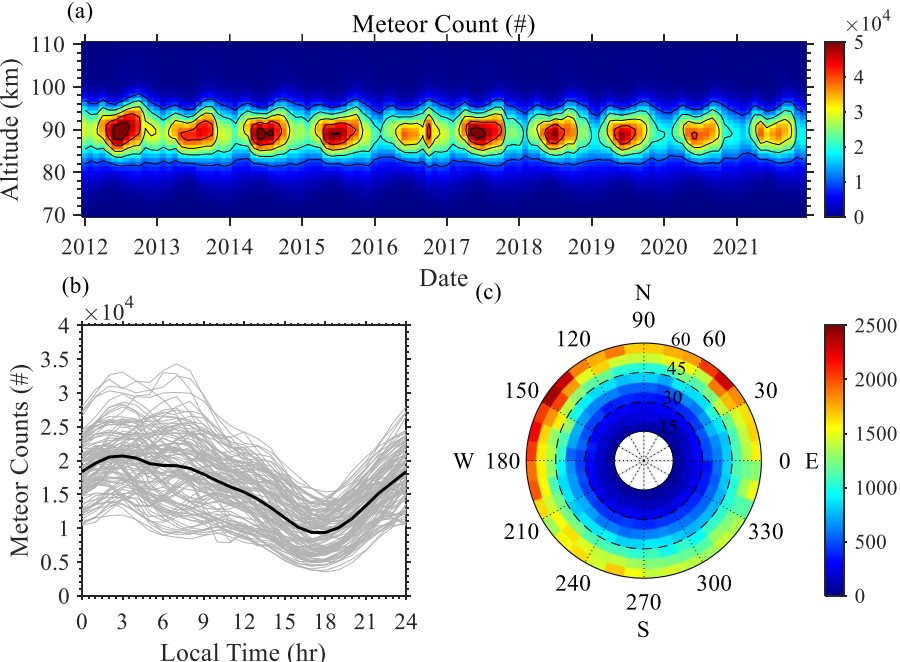

**Figure 1.** (**a**) Altitudinal variation of the selected meteor counts (#) in each month for the period between 2012 and 2021. (**b**) Local-time variation of the meteor counts in a 1-h bin for each month (grey). The black line is the average value. (**c**) Azimuth-zenith distribution of the meteor counts in January 2012.

Hocking's approach described a matrix equation between the detected radial velocity, angular information, and the GW momentum fluxes/variance, based on the minimization of the difference between the square of the radial velocity ($v_{rad}$) and the projected mean wind ($v_{pm}$). To solve the matrix equation, adequate data points are necessary. The method of composite day (CD) in one month is then used when detected meteor counts are inadequate for a single day, which is then widely adopted by the meteor radars [31]. However, the

CD method would induce the contamination of tides and other waves. To deal with this problem, the improvements made by Andrioli et al. [32] minimized the contamination by the atmospheric tides, the planetary waves, and the prevailing winds. They also suggested that such a contamination is more serious when semidiurnal tides are strong, which is the exact situation for Mohe [33]. Applying the modified composite day (MCD) method, Pramitha et al. [34] estimated the capability of three meteor radars over the tropical region, to retrieve the GW momentum fluxes. Tian et al. [30] used another similar but nonidentical method to reduce the influence from the day-to-day varied tides and planetary waves (PWs).

In the present work, we roughly follow the MCD method to reduce the impact of the tides and PWs. The detailed data processing is described as follows: (1) The data of $v_{rad}$ are divided into the 3-km height and 2-h bin with the stepping of 1-km and 1-h. The conventional CD method is used to estimate the GW momentum fluxes. Notably, if the deviation of $v_{pm}$ and $v_{rad}$ is larger than the three times standard deviations in each bin, we would exclude those data for basic quality control. If the data number is less than seven and 30 in each time/height bin, we will discard them in the estimation of mean wind and GW momentum fluxes, respectively. (2) Calculating the mean horizonal wind in each time/height bin for every day and fitting the tides and 2-day PWs in a 3-day window with a 1-day sliding. Then, the fitting tides and PWs are projected to the measured data points ($v_{tp}$) and replace $v_{rad}$. (3) Compositing the $v_{tp}$ in each month and inferring the artificial GW momentum fluxes, caused by the fitting tides and the PWs in each bin. Finally, subtracting the artificial GW momentum fluxes from those estimated in step (1). Figure 2a,b illustrates an example of the GW zonal and meridional momentum fluxes using the above processes (MCD method) in January 2012. The results infer that the tides and PWs would contaminate the estimated GW momentum fluxes by about 10%, which is in similar magnitude, shown in the work by Andrioli et al. [32]. In addition, the data processing also have the capability to retrieve the wind variances in reasonable magnitude in this work. Notably, the method also has a limitation in that if the data points are inadequate, the variance of the vertical wind ($w'^2$) will be estimated as negative values in mathematics, which is not reasonable in physics. We hope to focus on the GW momentum fluxes in this paper, so the wind variance is not displayed here.

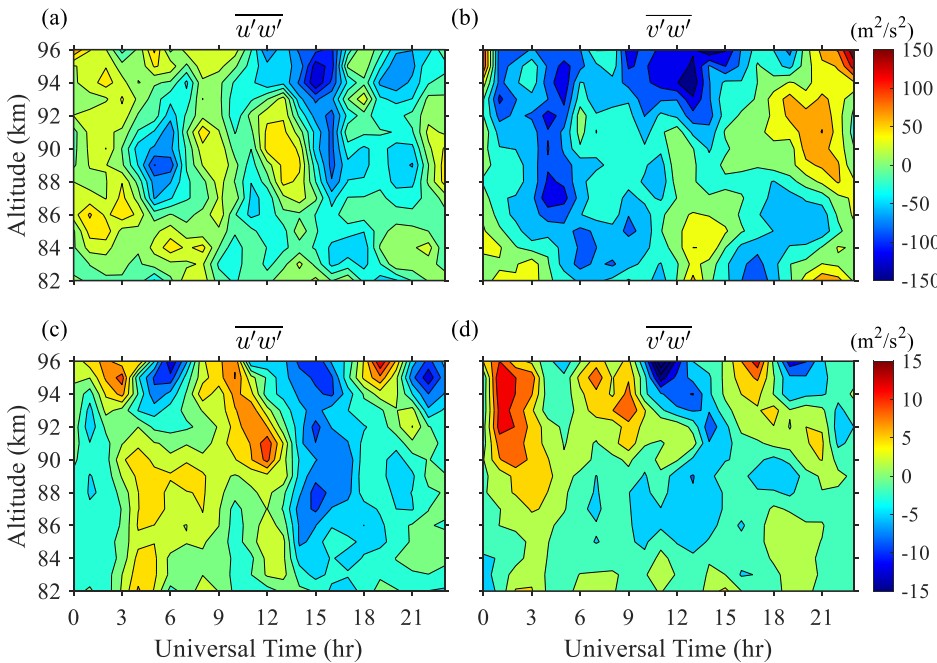

**Figure 2.** Altitude-time variation of the zonal (**a**,**c**) and meridional (**b**,**d**) momentum fluxes calculated by (**a**,**b**) the MCD method and (**c**,**d**) the compositing tides and the PWs during January 2012.

## 3. Results

The following work is to assess how well the MCD method is on estimating the GW momentum fluxes, by applying on the Mohe meteor radar. We adopt the test method proposed by Fritts et al. [18], which is then improved in the following works of Andrioli et al. [32] and Pramitha et al. [34]. The processes are summarized as follow: (1) constructing wind fields artificially to the actual angular information as the radar detecting. The pre-specified wind fields include the consideration of various GWs, together with mean winds, tides, and planetary waves. (2) As the measured radial velocity is replaced by the artificially constructed velocity, the MCD analysis then resolves GW momentum fluxes. (3) Comparing the resolved and prescribed GWs, we could present a quantitative evaluation on the accuracy and uncertainty. The pre-specified wind fields are calculated as the following Equations (1)–(4):

$$
\begin{aligned}
U(x,y,z,t) \quad &= U_M + U_{2D}(t)\sin(2\pi(t-\delta_{U2D})/T_{2D}) \\
&+ U_D(z,t)\sin(2\pi(t-\delta_{UD})/T_D) + U_{SD}(z,t)\sin(2\pi(t-\delta_{USD})/T_{SD}) \\
&+ \sum_{i=1}^{N} U_{GWi}(x,y,z,t)\sin(k_i x + l_i y + m_i z - 2\pi t/T_{GWi})
\end{aligned}
\tag{1}
$$

$$
\begin{aligned}
V(x,y,z,t) \quad &= V_M + V_{2D}(t)\sin(2\pi(t-\delta_{V2D})/T_{2D}) \\
&- V_D(z,t)\sin(2\pi(t-\delta_{VD})/T_D) + V_{SD}(z,t)\sin(2\pi(t-\delta_{VSD})/T_{SD}) \\
&+ \sum_{i=1}^{N} V_{GWi}(x,y,z,t)\sin(k_i x + l_i y + m_i z - 2\pi t/T_{GWi})
\end{aligned}
\tag{2}
$$

$$
W(x,y,z,t) = \sum_{i=1}^{N} W_{GWi}(x,y,z,t)\sin(k_i x + l_i y + m_i z - 2\pi t/T_{GWi})
\tag{3}
$$

$$
V_{rad} = U\sin\theta\cos\varphi + V\sin\theta\sin\varphi + W\cos\theta
\tag{4}
$$

The prescribed constructing wind, considered the zonal and meridional mean winds ($U_M$, $V_M$), diurnal and semidiurnal tides ($U_D$, $V_D$; $U_{SD}$, $V_{SD}$), 2-day PWs ($U_{2D}$, $V_{2D}$), and different $N$ kinds of GWs ($U_{GWi}$, $V_{GWi}$, $W_{GWi}$). ($k_i$, $l_i$, $m_i$) are the zonal, meridional, and vertical wavenumber of the $i$-kind GW, and $T_{GWi}$ describes the corresponding GW period. The symbols of $x$, $y$, and $z$ indicate the location of the detected meteors in east, north, and vertical direction taken by the radar as the coordinate origin. $\theta$ and $\varphi$ are the zenith and azimuth angles. $t$ is the universal time and ($\delta_{UD}$, $\delta_{USD}$, $\delta_{U2D}$; $\delta_{VD}$, $\delta_{VSD}$, $\delta_{V2D}$) indicate the phase of the diurnal tides, semidiurnal tides, and 2-day PWs in the zonal and meridional winds, respectively. ($T_D$, $T_{SD}$, $T_{2D}$) are the periods of the diurnal tides, semidiurnal tides, and 2-day PWs. The typical assignment of the parameters is listed in Table 1, in detail. Here, we performed four different cases, which represent different conditions of the GWs accompanied with other waves. As Mohe is the midlatitude location, the semidiurnal tides are set to be much stronger than those aforementioned in the previous studies. Here, an example of the prescribed wind fields with GWs within the interval of $\pm5$ min is shown in Figure 3. The example is corresponding to Case 2 and the angular information of the detected meteors are at the universal time of one o'clock (UT0100) and 10 min later (UT0110), both for the month of June 2012. Obvious wave structures could be seen in Figure 3a,d. As specified in Table 1, the zonal wind is dominant by a stationary GW with a zonal wavelength of 100 km (GW3), and a meridional wind is mainly a propagating GW with the period of 30 min (GW2). Figure 3a,b illustrates the dominant stationary GW zonal structure, and Figure 3c,d apparently shows the GW propagation in the meridional direction. Our aim is to extract the information of the GW momentum fluxes from such patterns.

**Table 1.** Parameters of the specified winds including the mean wind, tides, PWs, and GWs.

| Parameter (Unit) | Case 1 | Case 2 | Case 3 | Case 4 |
|---|---|---|---|---|
| $U_M, V_M$ (m/s) | 10, 10 | 10, 10 | 20, 40 | −20, −10 |
| $U_D, V_D$ (m/s) | 10, 10 | 10, 10 | 5, 5 | 10, 10 |
| $U_{SD}, V_{SD}$ (m/s) | 40, 40 | 40, 40 | $20 + 2(z − 80)\sin^2(\pi t/T_M)$ | 40, 40 |
| $\lambda_D, \lambda_{SD}$ (km) | 25, 50 | 25, 50 | 25, — | 25, 50 |
| $U_{2D}, V_{2D}$ (m/s) | $20 + 5R_0$ | 0, 0 | 0, 0 | 0, 0 |
| $(U, V, W)_{GW1}$ (m/s) | 10, 0, 5 | 10, 0, 5 | $20abs[\sin(2\pi t/T_M)]\sin(2\pi t/T_{SD}), 0,$ $−10abs[\sin(2\pi t/T_M)]\cos(2\pi t/T_{SD})$ | $40F_4(t)$ *, 0, $20F_4(t)$ * |
| $k_1, l_1, m_1$ (km$^{-1}$) | $2\pi/30, 0, 0$ | $2\pi/30, 0, 0$ | $2\pi/30, 0, 0$ | $2\pi/50, 0, 2\pi/15$ |
| $(U, V, W)_{GW2}$ (m/s) | 0, 20, 2 | 0, 20, 2 | $0, 20abs[\sin(2\pi t/T_M)]\sin(2\pi t/T_{SD}),$ $5 abs[\sin(2\pi t/T_M)]\cos(2\pi t/T_{SD})$ | $0, 30G_4(t)$ *, $10G_4(t)$ * |
| $k_2, l_2, m_2$ (km$^{-1}$) | $0, 2\pi/50, 0$ | $0, 2\pi/50, 0$ | $0, 2\pi/50, 0$ | $0, 2\pi/100, 2\pi/20$ |
| $(U, V, W)_{GW3}$ (m/s) | — | 20, 0, −10 | 20, 0, −10 | — |
| $k_3, l_3, m_3$ (km$^{-1}$) | — | $2\pi/100, 0, 0$ | $2\pi/100, 0, 0$ | — |
| $(U, V, W)_{GW4}$ (m/s) | — | 0, 10, 2 | 0, 10, 2 | — |
| $k_4, l_4, m_4$ (km$^{-1}$) | — | $0, 2\pi/80, 0$ | $0, 2\pi/80, 0$ | — |
| TGW1, TGW2 | 20, 30 | 20, 30 | 20, 30 | 20, 30 |
| TGW3, TGW4 (min) | —, — | ∞, ∞ | ∞, ∞ | —, — |
| $<u'w'>_{mean}$ (m$^2$/s$^2$) | 25 | −75 | −100 | 50 |
| $<v'w'>_{mean}$ (m$^2$/s$^2$) | 20 | 30 | 10 | 20 |

* mean GW momentum fluxes for each case is shown at the bottom, $F_4(t) = 1$ ($t = 0$–3 hr + $21R_1$ hr) and $F_4(t) = 0$ otherwise, and $G_4(t) = 1$ ($t = 0$–4 hr + $20R_2$ hr) and $G_4(t) = 0$ otherwise, with $R_1$ and $R_2$ random variables between 0 and 1 chosen separately for each day of the test month.

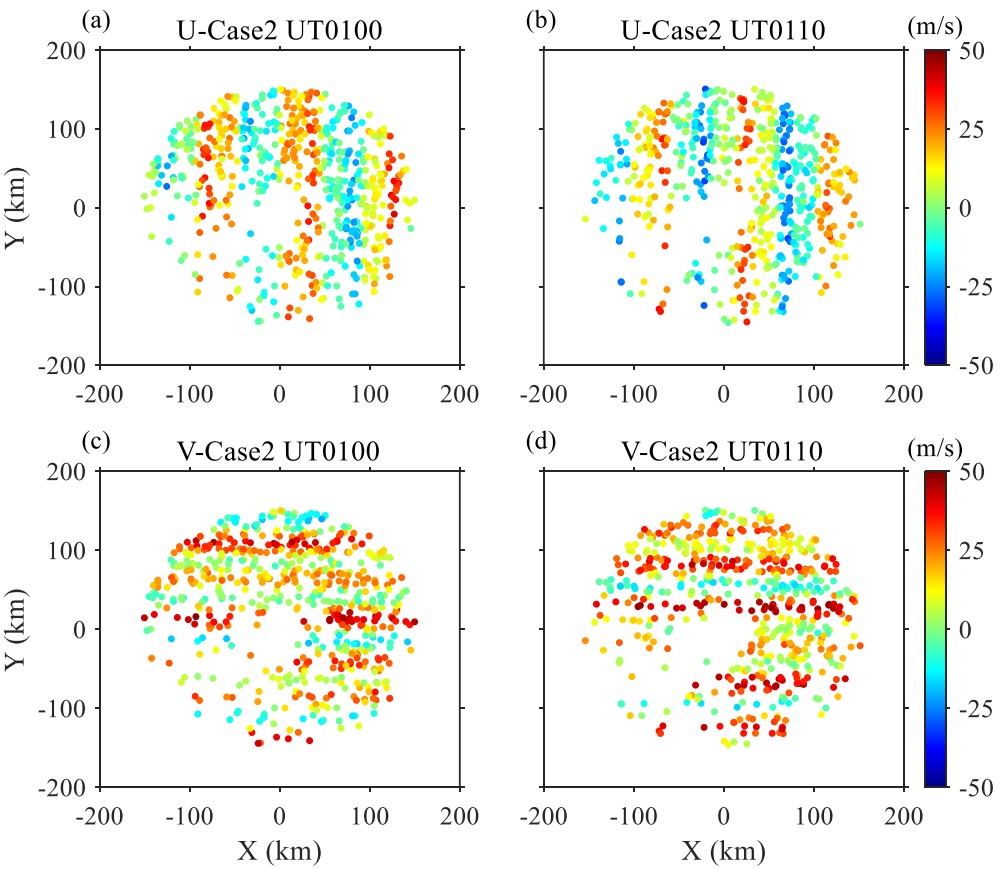

**Figure 3.** Example of the artificial prescribed constructing zonal (**a,b**) and meridional (**c,d**) winds with GWs for Case 2 at UT0100 and UT0110 within an interval of ±5 min at heights of 88.5–91.5 km in June 2012.

### 3.1. Case 1

Case 1 represents two high frequency GWs with only a zonal and meridional variation. The periods of the two GWs are 20 and 30 min, which is less than the 2-h time bin. The wavelengths are set to be 30 and 50 km in the zonal and meridional direction, respectively. The two GW amplitudes in the zonal, meridional, and vertical are specified to be (10, 0, 5) m/s and (0, 20, −2) m/s. Hence, the GW zonal and meridional momentum fluxes are calculated to be 25 m$^2$/s$^2$ and 50 m$^2$/s$^2$. The background mean wind, diurnal and semidiurnal tides are given as the typical values. Two-day waves with amplitudes of about 20 m/s in both the zonal and meridional direction are also concerned in this case. Small random fluctuations in the 2-day waves are also considered. Using the detected angular information by the Mohe meteor radar in July 2012, when the meteor counts are large enough, we could estimate the accuracy of the MCD method. Figure 4a shows the differences between the estimated tides and the prescribed ones. Both the diurnal and semidiurnal tides are estimated with an error around 5–10% with an insignificant altitudinal variation. The estimated GW momentum fluxes are illustrated in Figure 4b, with the comparison of the specified ones. Below ~94 km, the MCD method gives a reasonable estimation on both $u\prime w\prime$ and $v\prime w\prime$ with the error of less than 5 m$^2$/s$^2$. As the altitude increases to 96 km, the estimated $u\prime w\prime$ has over a 10 m$^2$/s$^2$ error.

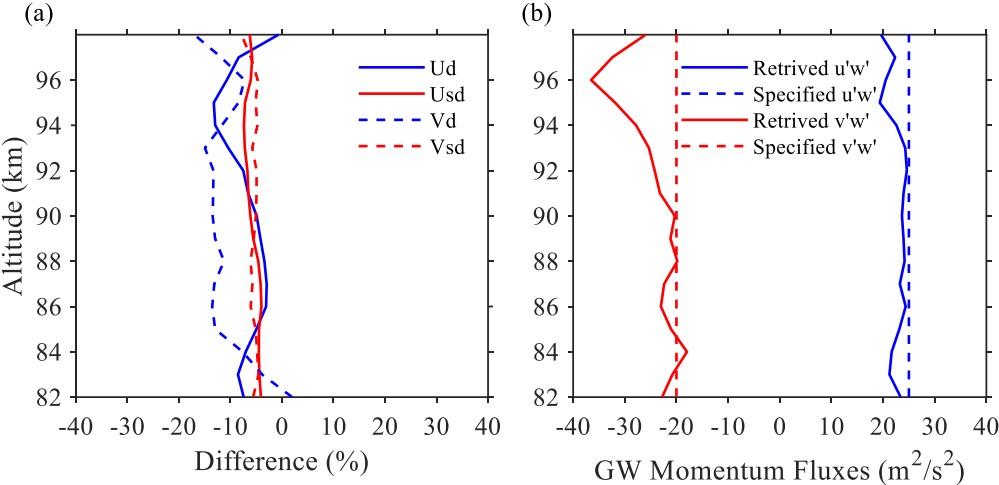

**Figure 4.** (**a**) Differences of the amplitudes of the diurnal (red lines) and semidiurnal (blue lines) tides, between the estimated ones and the specified ones for case 1, using the angular information detected in July 2012. The solid and dashed lines are for the zonal and meridional direction, respectively (**b**).

### 3.2. Case 2

Case 2 considers another two stationary GWs with prescribed horizontal wavelengths of 100 and 80 km, and the 2-day PW is excluded. Figure 5 illustrates the estimating tides and the GW momentum fluxes. Although the estimated diurnal tides have a slightly larger difference with the specified ones, than that in Case 1, the discrepancy is still less than ~20%. The estimated $u\prime w\prime$ and $v\prime w\prime$ agree well with the prescribed, and the error at a high altitude is still at an acceptable extent. The result indicates that the MCD method has a good performance in estimating the GW momentum fluxes from the mixed GWs, and the 2-day PW is likely to introduce an additional error in the GW estimation.

### 3.3. Case 3

Case 3 includes a 10-day wave modulation on the semidiurnal tides and an increasing amplitude with altitude. The amplitudes of two GWs are also considered to be modulated by the 10-day wave and semidiurnal tides. The detail of the two specified GW waves is listed in Table 1. Other two stationary GWs, the same as Case 2, are also included, which should represent a more realistic atmospheric condition. As shown in Figure 6a, the error of the estimated diurnal tides in Case 3 has a similar magnitude with Case 2 (within ±20%). The estimated semidiurnal tides agree well with the specific ones, for which the differences

are ~5%. As for the estimated $u\prime w\prime$ and $v\prime w\prime$, both of them show slight discrepancies with the specified $u\prime w\prime$ and $v\prime w\prime$, especially below ~94 km (Figure 6b). The results confirm the capability of the Mohe meteor radar in estimating the GW momentum fluxes using the MCD method in a complex atmospheric condition.

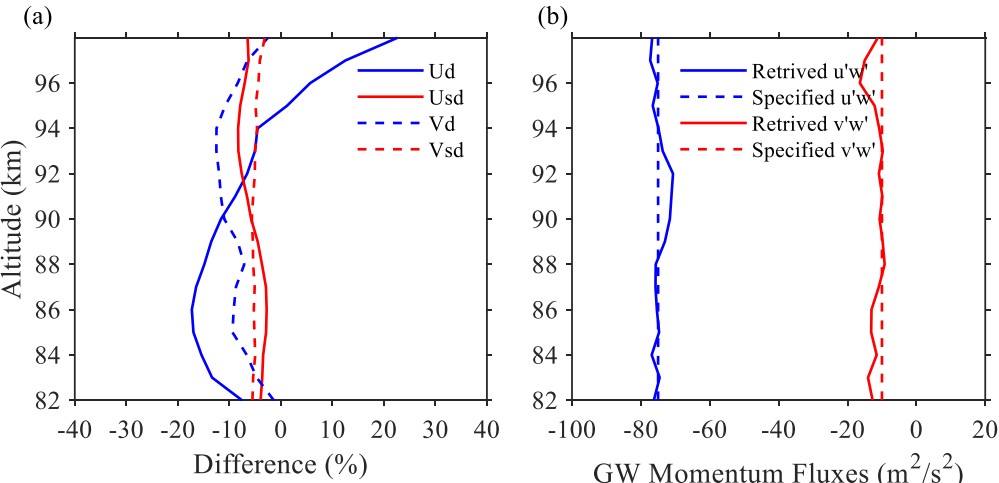

**Figure 5.** (**a**) Differences of the amplitudes of the diurnal (red lines) and semidiurnal (blue lines) tides, between the estimated ones and the specified ones for case 2, using the angular information detected in July 2012. The solid and dashed lines are for the zonal and meridional direction, respectively (**b**).

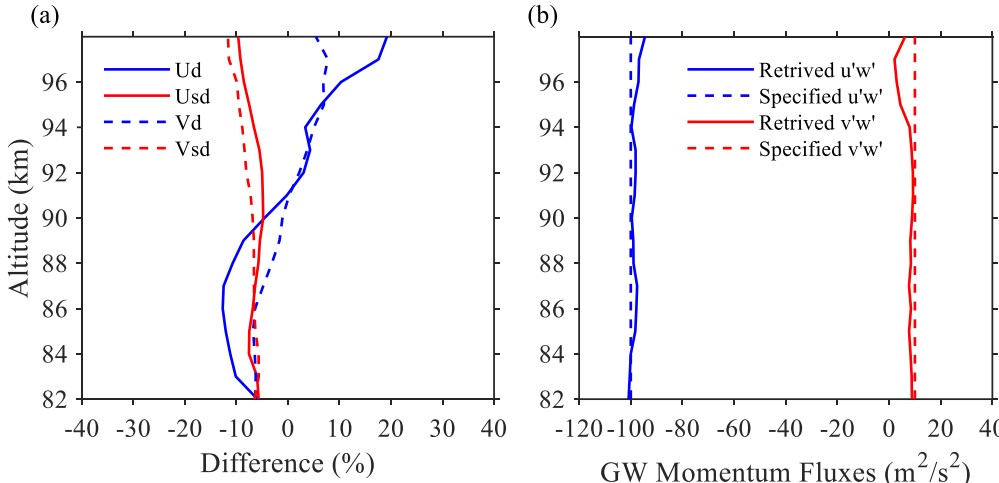

**Figure 6.** (**a**) Differences of the amplitudes of the diurnal (red lines) and semidiurnal (blue lines) tides, between the estimated ones and the specified ones for case 3, using the angular information detected in July 2012. The solid and dashed lines are for the zonal and meridional direction, respectively (**b**).

### 3.4. Case 4

Case 4 is for the situation considering intermittent GW activity. We introduce two randomly occurring GWs in constructing the prescribed winds, which exist about 3 and 4 h with a period of 30 and 20 min. As shown in Table 1, the introduced intermittent GW1 and GW2 are described by the random functions of $F_4$ and $G_4$, which mean that the GW1 and GW2 would occur randomly in one day (24 h) and last 3 and 4 h, respectively. In the rest of 20 and 21 h, the GW1 and GW2 do not exist. The horizontal and vertical wavelength is 50 km and 15 km for GW1, and 100 km and 20 km for GW2. Notably, as the GWs are randomly occurring in this case, the data quality control about the standard deviation should exclude the actual GW, rather than the outliers. Thus, this step is skipped in Case 4. The comparison results of the tides and the GW momentum fluxes are displayed in Figure 7. Both diurnal and semidiurnal tides are reconstructed reasonably, which have a small discrepancy with

the specific tides (~5%). The differences between the estimated and specified $u\prime w\prime$ and $v\prime w\prime$ are less than ~2 m$^2$/s$^2$. The results inferred that the Mohe meteor radar also has a good performance in estimating the momentum fluxes of the intermittent GWs.

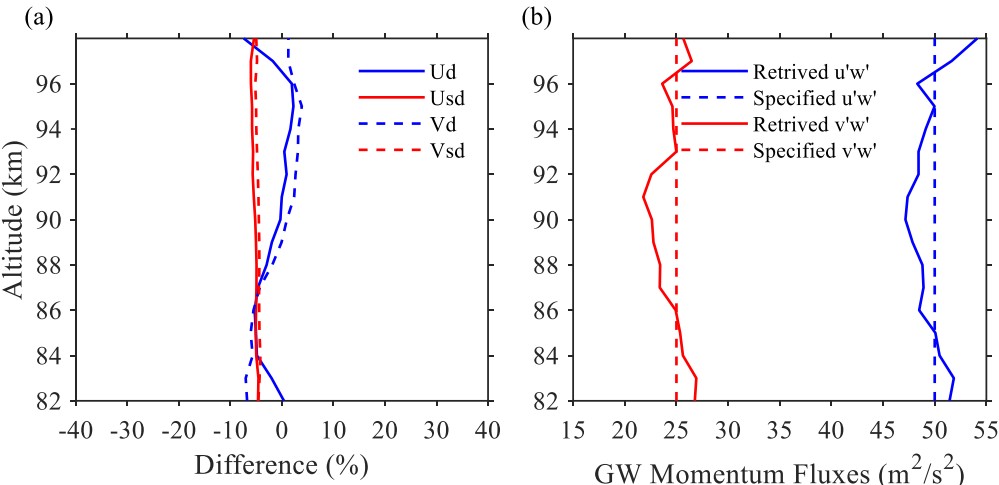

**Figure 7.** (**a**) Differences of the amplitudes of the diurnal (red lines) and semidiurnal (blue lines) tides, between the estimated ones and the specified ones for case 4, using the angular information detected in July 2012. The solid and dashed lines are for the zonal and meridional direction, respectively (**b**).

The above analysis shows that a plausible GW estimation could be given by the Mohe meteor radar observation. We further examine the four cases throughout each month during 2012–2021. Figure 8a–d shows the comparison between the estimated and the specific $u\prime w\prime$ and $v'w'$, every month. The results infer that the MCD method applied to the Mohe meteor radar observations give a reasonable estimation within the heights of 82–94 km. Above 94 km, the bias of the estimated GW momentum fluxes for all of the four cases clearly becomes large. Although the GWs are prescribed as identical, the estimated GWs have a clear uncertainty for each month, which should arise from the different angular information of the detected meteors. The uncertainty is about 2 m$^2$/s$^2$ below 94 km, but increases rapidly from that altitude (Figure 8e–h). The estimation uncertainty of the meridional momentum fluxes is slightly larger than that of the zonal momentum fluxes, except for Case 1. The results indicate that the Mohe meteor radar could be used to study the variability of both the zonal and meridional GW momentum flux larger than 2 m$^2$/s$^2$. If the variability signal is less than 2 m$^2$/s$^2$, the result might be spurious and need to be reassessed.

Applying the MCD analysis to the decadal continuous observation data (2012–2021) from the Mohe meteor radar, we also estimate the realistic mesopause GW momentum fluxes. Both monthly $u\prime w\prime$ and $v\prime w\prime$ have obvious variations in time and altitude (Figure 9a,b). The zonal momentum fluxes are generally positively strong in summer (~20–30 m$^2$/s$^2$) and become weak at around zero in winter. The meridional momentum fluxes are also strong in summer, but are secondarily positively peak in winter, in some years. In addition, there are apparent interannual variability in both zonal and meridional momentum fluxes. The zonal momentum fluxes are extremely strong in 2012 and weak in 2013, then gradually strengthen from 2014 to 2017, and have become weak again since 2018. As for the meridional momentum fluxes, the magnitude in 2012 are also stronger than in other years. As the altitude increases, the zonal (meridional) momentum fluxes in summer change from eastward (northward) to westward (southward). The negative vertical shear of the momentum fluxes is expected to be deposited at an eastward and northward acceleration in the zonal and meridional winds, respectively. It is exactly shown in Figure 9c,d, that presents the averaged seasonal climatology of the GW momentum fluxes and the mean wind during the 10 years. The annual cycle and altitudinal variation of the momentum fluxes support the selective filtering mechanism (Fritts and Alexander, 2003). The westward mean winds at ~82 km in summer allow the eastward GWs to propagate to the higher

altitude and drag the mean wind eastwards. The eastward mean winds at a higher altitude would filter the eastward GW in turn, and the *u′w′* would become small. These results explain how the GWs interact with the mean winds over Mohe. In addition, Figure 9e gives the monthly variation of the geomagnetic and solar activities. The geomagnetic activity manifests to not be related to the interannual variation of the GW variability. As the solar activity decreased in 2017–2020, the eastward GW momentum fluxes became stronger than in 2013–2014, the period with a higher solar activity. Cullens et al. [35] used numerical simulations to examine such an anticorrelation, which should arise from the impact of solar activity on the general circulation around the mesopause. However, the relationship between the solar activity and the GW momentum fluxes is obviously not valid for the year 2012, which has stronger eastward GW momentum fluxes with a high solar activity. More evidence with the longer observations in the future is necessary to examine the reliability of such a relationship.

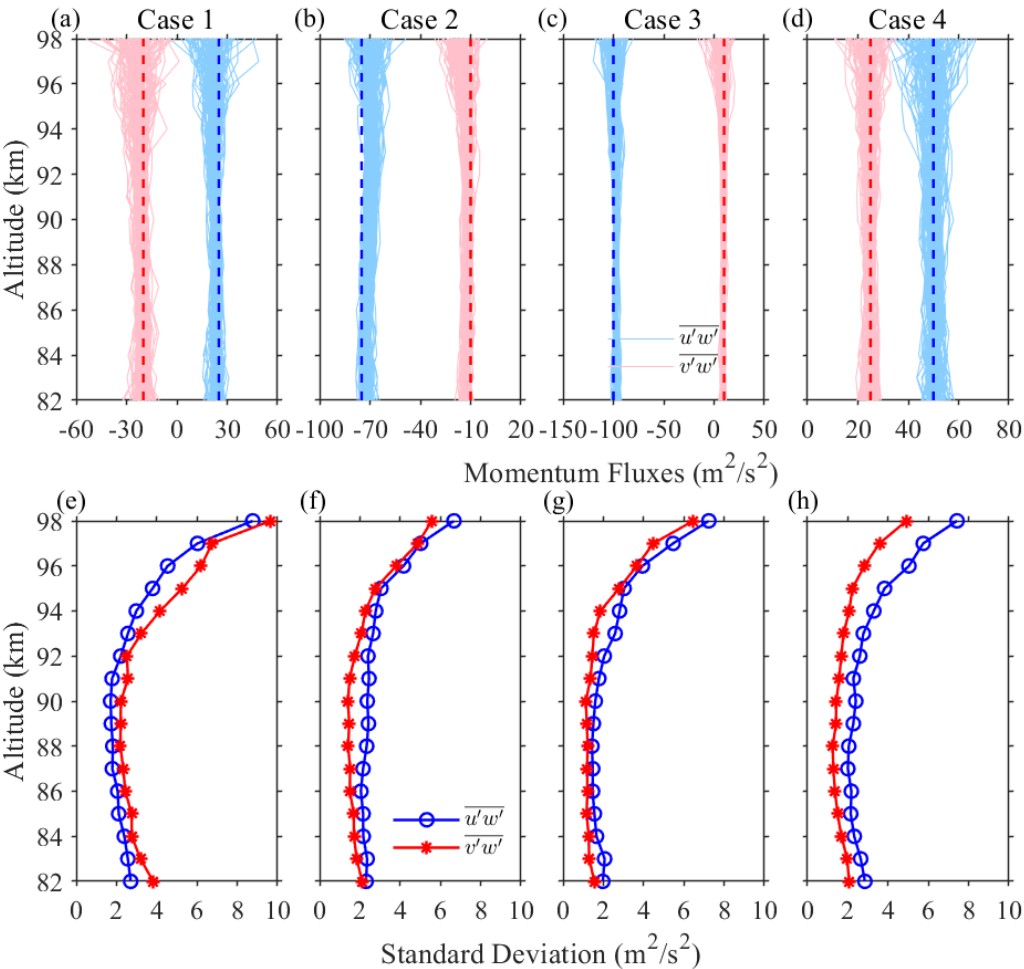

**Figure 8.** (**a**–**d**) Estimated zonal (sky-blue lines) and meridional (pink lines) GW momentum fluxes using the MCD method in each month, for the period between 2012 and 2021. The specific mean GW momentum fluxes are plotted in blue (red) dashed lines. (**e**–**h**) The altitudinal variations in the standard deviation of the vertical fluxes of the zonal and meridional GW momentum are plotted in blue solid-circles and red solid-asterisk lines, respectively. Columns from left to right are for Cases 1–4.

To examine the interannual variability, we deseasonalize the estimated GW zonal and meridional momentum fluxes with a detrend analysis, as shown in Figure 10a,b, respectively. Most of deseasonalized *u′w′* and *v′w′* fluctuated from −15 to 15 m$^2$/s$^2$. A significant enhancement of *u′w′* and *v′w′* is found at 82 km in 2012, and a depression is found in 2013. Analysis of the results by Lomb-Scargle is shown in Figure 11. Results show

that there is no obvious QBO-like signal, as shown by de Wit et al. [6]. The signal with long periods over 5 years needs a longer time series to confirm the reliability in the future.

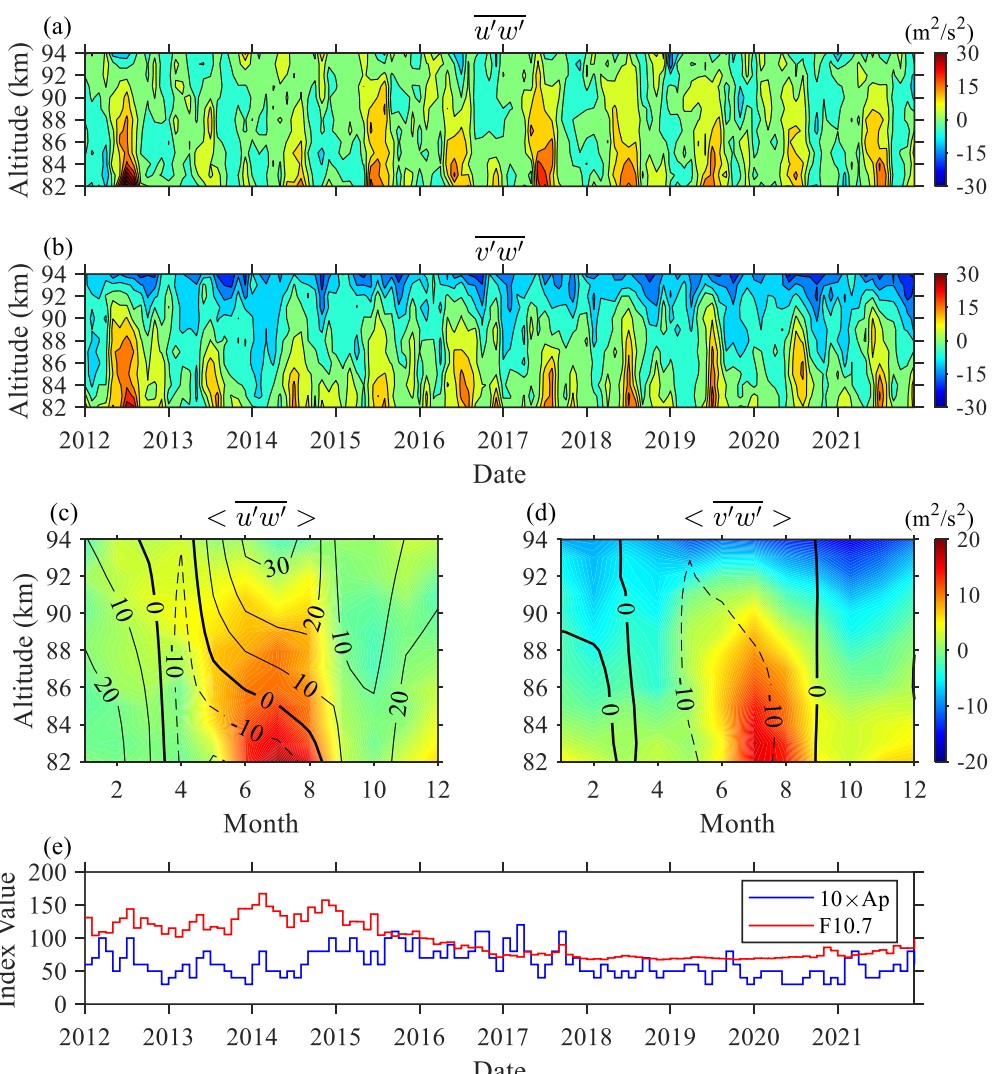

**Figure 9.** Estimated (**a**) zonal and (**b**) meridional GW momentum fluxes in different altitudes, for the period between 2012 and 2021. Contours (**c,d**) are the averaged results for the zonal and meridional GW momentum fluxes, respectively. Thin black solid (dash) lines indicate the decadal averaged monthly-mean winds with positive (negative) values. Thick black lines are zero winds. (**e**) Monthly variations of the geomagnetic (Ap, blue line) and solar flux (F10.7, red line) index.

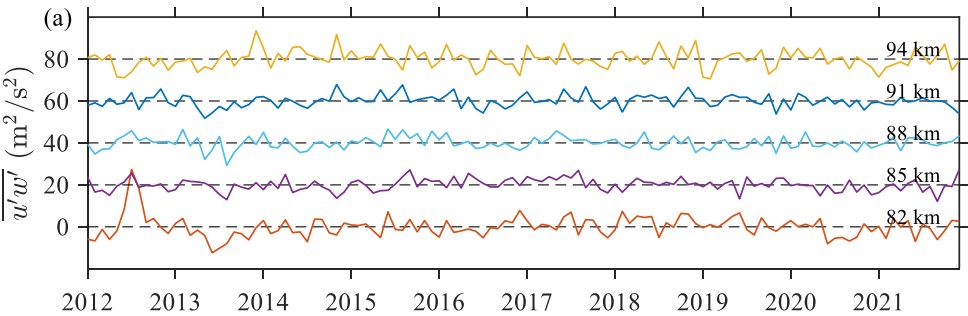

**Figure 10.** *Cont.*

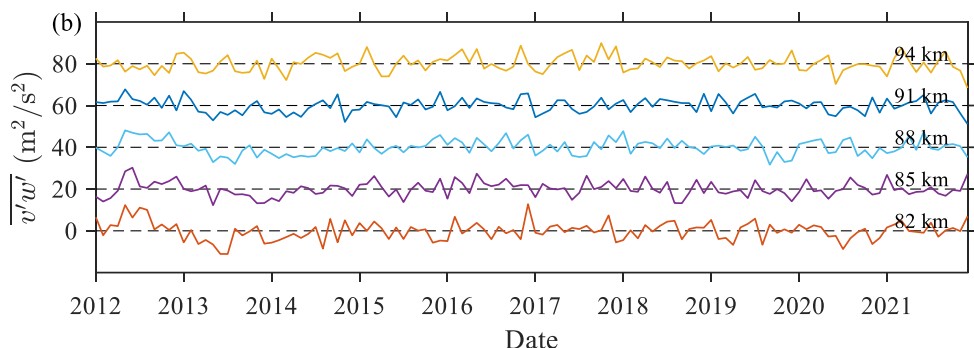

**Figure 10.** Time series of the detrended and deseasonalized (**a**) zonal and (**b**) meridional momentum fluxes at 82, 85, 87, 88, and 94 km, stacking with an interval of 20 m$^2$/s$^2$. The dashed lines are for zeros for each height.

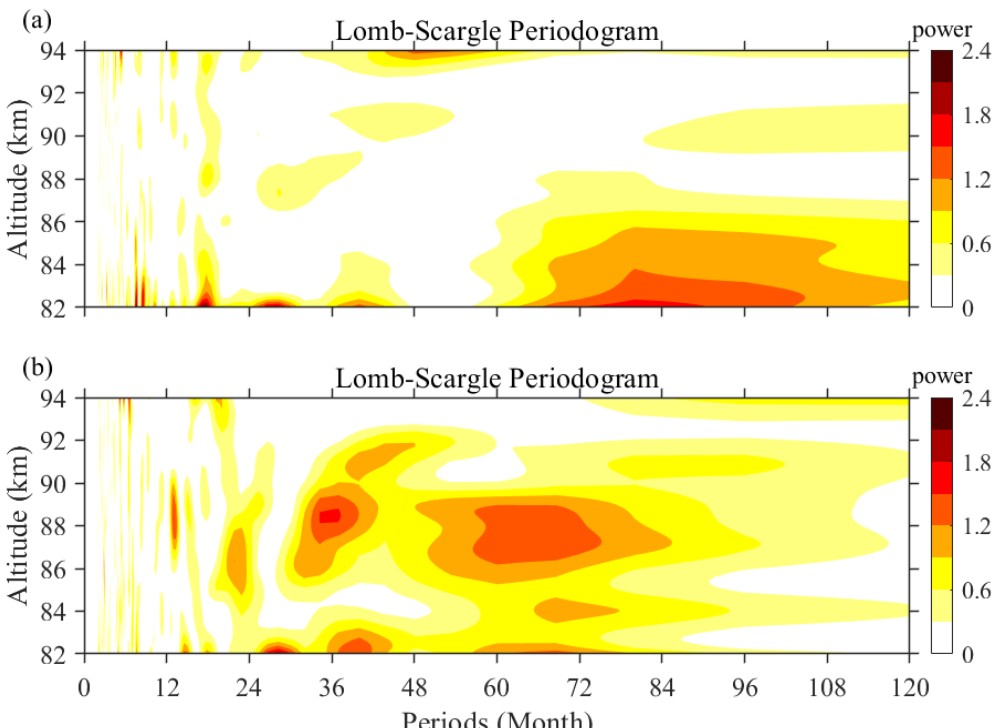

**Figure 11.** Lomb–Scargle Periodogram of the deseasonalized (**a**) zonal and (**b**) meridional momentum fluxes.

## 4. Discussion

The Mohe meteor radar has been operating uninterruptedly for the past decade. Numerous scientific studies have been published to discuss the wind or tidal variabilities detected by the radar, with broad ranging timescales of day-to-day [36], seasonal [33], intraseasonal [37], semiannual, and annual [38]. In this work, we aimed to evaluate the capability of the Mohe meteor radar in estimating the GW momentum fluxes and then discuss the interannual variability. Although Hocking's method has also been used for other meteor radars worldwide, the estimated GW momentum fluxes depend on each radar's performance. For instance, Pramitha et al. [34] analyzed three meteor radars over the India region, and found that the Thumba and Tirupati stations have a better performance than the Kototabang station. de Wit et al. [6] discussed the interannual variability in Argentina by 7-yr interval data while only 48-months are available. Our work shows that the Mohe meteor radar has a good performance on estimating the GW momentum fluxes uninterruptedly for the past decade (Figure 7).

The estimated GW momentum fluxes are also in reasonable value with comparisons of previous studies. Ma et al. [38] estimated the GW forcing from the mean winds over the Mohe, based on the assumption that the primary force balance in the mesosphere and lower thermosphere (MLT) is between the zonal mean GW force and the Coriolis force. Their results also show that the zonal GW force has a similar seasonal variation to our results of the zonal GW momentum fluxes (Figure 9c). As the air density is another factor determining the GW force, we roughly estimate the GW force by implementing the air density data from the MSIS model. The maximum GW force in summer is about (150–200 $m^2/s^2$), which is in the same magnitude as Ma's work (not shown). Their result supports our result giving a reasonable estimation. In addition, Jia et al. [28] also presented the estimation of the GW momentum fluxes over Mohe, while they did not give the reliable analysis of their method and displayed the interannual variability. Their result is smaller than our work, and we believe that the discrepancies come from one of data selections they used. In the data processing, they exclude the data that have differences exceeding 25 m/s with the projected mean radial wind. We tried to include this step in our processes, and all four cases in Section 3 show underestimated GW momentum fluxes (not shown).

The variability of the GW momentum fluxes is determined by the wave sources and background wind, during their upward propagation. However, there is rare knowledge about the GW source variability over Mohe. Figure 3f of Ern et al. [20], should be a reference map showing that Mohe is the location of the moderate GW momentum fluxes in the stratosphere. Figure 12 displayed the background wind from the ERA-5 re-analysis data and the meteor radar itself. The figure shows there is a strong westward jet at ~50–70 km in summer that would allow the eastward GW propagating into the MLT region. The maximum of the westward jet and the mesopause GW momentum fluxes are consistently occurring in July, slightly later than the June solstice. We suggested that the mesospheric circulation is determined by not only the radiative effect, but also by the dynamical processes (e.g., eddy forcing). As reviewed by Baldwin et al. [39], the mesospheric QBO is one of the dominant modes in wind oscillation, although the magnitude is not strong as in the tropical region. de Wit et al. [6] also reported a QBO modulation on the GWs at a midlatitude of the southern hemisphere. However, in this study, the QBO-like signal in the GW momentum fluxes is insignificant, especially compared with the uncertainty by the estimating method. Some other studies [40] also suggested that the mesospheric QBO is not as strong as Baldwin et al. [39] declared. In addition, the mesospheric GW momentum fluxes could also be modulated by some other potential sources on the interannual time scale. Cullens et al. [35] found that, as the solar activity increases, the GW force around the mesopause generally decreases, by analyzing the WACCM simulation. ENSO is another potential factor that suggests a 70% interannual variability of the stratospheric gravity wave in the South Pacific, regressed on the ENSO index, based on high-resolution AIRS observations [41]. Thus, the underlying physics mechanism and the dominant factor of the GW momentum fluxes over Mohe (Figure 9) should be complicated and need further analysis in the future.

## 5. Conclusions

This paper firstly evaluates the capability of estimating the GW momentum fluxes by the Mohe meteor radar, by designing several experiments. Results confirm that the Mohe meteor radar has the capability to estimate the GW momentum fluxes properly in the height of 82–94 km with errors of less than 5 $m^2/s^2$ when the meteor counts are adequate. The estimated uncertainty caused by the different angular information of the detected meteor in each month is about 2 $m^2/s^2$. The estimated GW momentum fluxes during the decade of 2012–2021 shows a clear seasonal and altitudinal variation, which is consistent with the selective filtering mechanism. The interannual variation shows a significant enhancement in the summer of 2012 and a depression in the summer of 2013. Slight changes are also found in other years. However, there is no apparent QBO-like signal in the Mohe GW momentum fluxes.

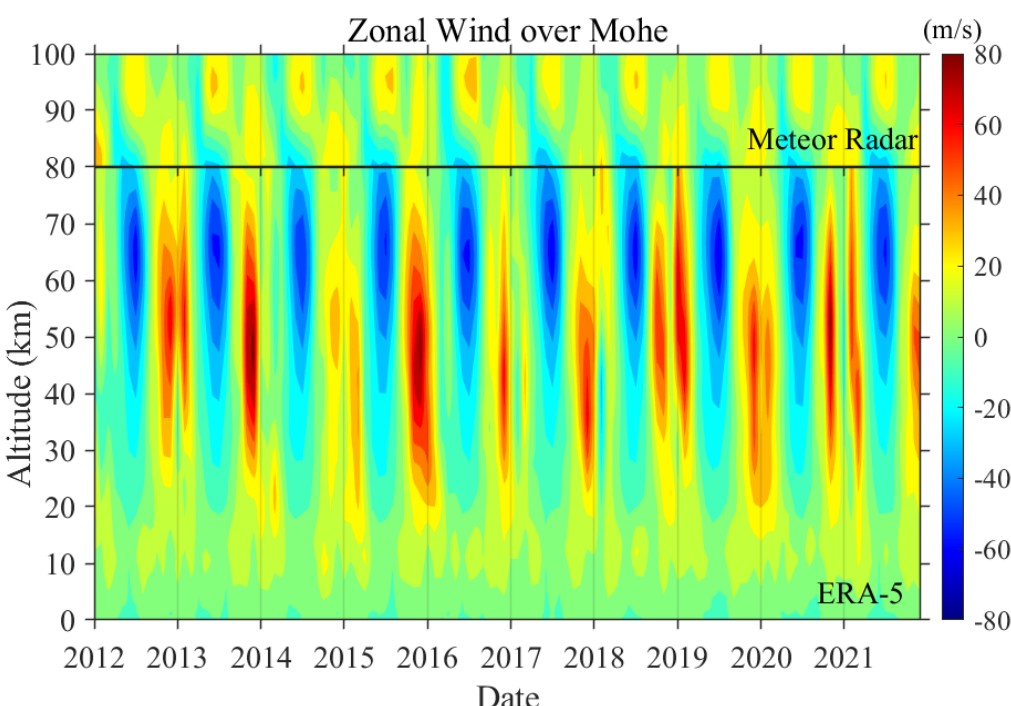

**Figure 12.** Monthly mean zonal wind over Mohe, from 2012 until 2021. The wind profile below 80 km is provided by the ERA-5 re-analysis data.

**Author Contributions:** Conceptualization, X.Z., F.D. and Z.R.; Data curation, X.Z. and Y.J.; Funding acquisition, X.Y. and L.L.; Investigation, X.Z.; Methodology, X.Z. and Y.Y.; Project administration, X.Y.; Supervision, X.Y. and F.D.; Visualization, X.Z. and H.Y.; Writing–original draft, X.Z.; Writing–review & editing, X.Z., X.Y., Y.Y., Z.R. and Y.J. All authors have read and agreed to the published version of the manuscript.

**Funding:** This research was funded by the Project of Stable Support for Youth Team in Basic Research Field, CAS (YSBR-018), the Chinese Meridian Project, and the B-type Strategic Priority Program of CAS (Grant No. XDB41000000). X.Z. thanks the China Postdoctoral Science Foundation (2021M703194).

**Data Availability Statement:** Meteor radar data were provided by the Beijing National Observatory of Space Environment, Institute of Geology and Geophysics Chinese Academy of Sciences through the Geophysics center, the National Earth System Science Data Center (http://wdc.geophys.ac.cn/dbList.asp?dType==MetPublish (accessed on 20 April 2022)). The website registration is open and free to the public. Ap and F10.7 data were downloaded from the National Geophysical Data Center (https://www.ngdc.noaa.gov (accessed on 2 May 2022)). ERA-5 data can be downloaded for the public from the website of https://www.ecmwf.int/en/forecasts/datasets/reanalysis-datasets/era5 (accessed on 8 June 2022).

**Acknowledgments:** We thank Mingjiao Jia and Jianfei Wu for their valuable help and suggestions.

**Conflicts of Interest:** The authors declare no conflict of interest.

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
