# Peer review of "Decadal Continuous Meteor-Radar Estimation of the Mesopause Gravity Wave Momentum Fluxes over Mohe: Capability Evaluation and Interannual Variation"

_remotesensing, doi:10.3390/rs14225729_

Round 1
Reviewer 1 Report (Previous Reviewer 1)
The revised manuscript has improved. However, the reviewer insists that a short paragraph about the occurrence of negative values of w^2 is added. This is important as it points out that the method has its limitations. To discuss such limitations is more relevant to the scientific community than sometimes the specific values of some observations.
Author Response
Point 1: The revised manuscript has improved. However, the reviewer insists that a short paragraph about the occurrence of negative values of w^2 is added. This is important as it points out that the method has its limitations. To discuss such limitations is more relevant to the scientific community than sometimes the specific values of some observations.
Response 1: Thanks for the comment on the method. According to the reviewer's suggestion, we have pointed out the method limitation as follow:
Notably, the method also has a limitation in that if the data points are inadequate, the variance of vertical wind (w'^2) will be estimated as negative values in mathematics, which is not reasonable in physics.
Reviewer 2 Report (Previous Reviewer 3)
The paper entitled “Decadal continuous meteor-radar estimation of mesopause gravity wave momentum fluxes over Mohe (53.5°N, 122.3°E): capability evaluation and interannual variation” by Zhou et al., calculated gravity wave momentum fluxes using decadal continuous observations by Mohe (53.5°N, 122.3°E) meteor radar. They evaluated the capability of meteor radar in retrieving gravity wave momentum fluxes using Hocking’s method in the MCD way. Each case of simulated and retrieved gravity wave momentum fluxes are matching well. The paper is organized well and I recommend a minor revision.
Comments:
-
Line 209…..Please explain what is UT0100 and UT0110
-
Line 323….. Vertical share??
-
Line 332…. Geomagnetic activity and solar activity during the given period are given in figure 9e, please discuss how these influence the GW momentum flux. LSP shows solar effect at the altitudes ranges 82-90km.
-
Line 358…. With board ranging???
Author Response
We thank the reviewer's suggestions.
Point 1: Line 209…..Please explain what is UT0100 and UT0110
Response 1: Thanks for the comment. We have revised the manuscript as "...are at the universal time of one o’clock (UT0100) and 10 minutes later (UT0110)..."
Point 2: Line 323….. Vertical share??
Response 2: Sorry for the typo. It is "shear", and we have corrected it.
Point 3: Line 332…. Geomagnetic activity and solar activity during the given period are given in figure 9e, please discuss how these influence the GW momentum flux. LSP shows solar effect at the altitudes ranges 82-90km.
Response 3: Thanks for the comment. We have added some sentences to describe the relationship.
“The geomagnetic activity manifests to be not related to the interannual variation of GW variability. As the solar activity decreases in 2017–2020, the eastward GW momentum fluxes become stronger than in 2013–2014, the period with higher solar activity. Cullens et al. (2016) used numerical simulation to examine such an anticorrelation, which should arise from the impact of solar activity on the general circulation around the mesopause. However, the relationship between solar activity and GW momentum fluxes is obviously not valid for the year 2012, which has stronger eastward GW momentum fluxes with high solar activity. More evidence with longer observations in the future is necessary to examine the reliability of such a relationship. ”
Point 4: Line 358…. With board ranging???
Response 4: Thanks for pointing out the typo on the "broad". We have corrected it.
Reviewer 3 Report (Previous Reviewer 5)
It is of interesting to develop ways to distinguish acoustic-gravity waves of natural and artificial origin.
Author Response
Thanks for the review.
Reviewer 4 Report (Previous Reviewer 2)
The author has improved the paper well compared with the last round of the review. Now the discussion is pretty sufficient to show the underlying mechanism of the GW seasonal variations at Mohe. I appreciate the detailed discussion about the QBO of the GW momentum flux.
Line 385-386 I do not think Liu 2019 is similar work to your work. That paper is a numerical modeling paper, not observation paper.
Line 391 showing that XXX
Line 393 figure shows that
Author Response
We appreciate the review. Here are the point-to-point responses.
Point 1: Line 385-386 I do not think Liu 2019 is similar work to your work. That paper is a numerical modeling paper, not observation paper.
Response 1: Thanks for the suggestion. We have removed the citation and deleted relevant content.
Point 2: Line 391 showing that XXX
Response 2: Thanks. We have corrected this grammar mistake.
Point 3: Line 393 figure shows that
Response 3: Thanks for pointing out the mistake, and we have corrected it.
This manuscript is a resubmission of an earlier submission. The following is a list of the peer review reports and author responses from that submission.
Round 1
Reviewer 1 Report
Review on:
Decadal continuous meteor-radar estimation of mesopause gravity wave momentum fluxes over Mohe (53.5°N, 122.3°E): capability evaluation and interannual variation
By
Xu Zhou, Xinan Yue, Libo Liu, You Yu, Feng Ding, Zhipeng Ren, Yuyan Jin and Hanlin Yin
General Comment:
The manuscript compares two methods to obtain momentum fluxes by applying the algorithm proposed by Hocking et al., 2005. The method is compared to synthetic data, which is almost identical to the cases discussed in Fritts et al., 2010. The differences between the cases are almost negligible.
In summary, the paper presents results based on 10 years of data collected with the Mohe meteor radar.
Detailed comments:
Figure 2:
This figure deserves a bit more controversial discussion as the momentfluxes v’w’ don’t agree at all between the two methods? Why is that the case? For the other component, this is less obvious, but still, there are substantial differences.
Line 151:
Improved appears to be not the best choice. The method of obtaining momentum fluxes was not changed. These papers only applied synthetic data on how the data needs to be prepared to avoid biases. However, the key conclusion was always that tides and planetary waves need to be carefully removed from the observations? How was this achieved in the present work? The authors state that the semidiurnal tides are significant above Mohe. How did they account for that in the analysis? They need to add some sentences what they did to deal with this aspect.
Wind variances:
Hocking’s method also obtains solutions for the wind variances. How many negative values are obtained from the u2,v2,w2?
Conclusions:
The conclusions should be somehow become more substantial. Right now there is only the statement that 10 years of data was analyzed, but no real conclusion is reached in a geophysical context. Why one should analyze 10 years of data. Are the momentum fluxes as expected according to other observations ? Are the results in agreement to model data? Please add a paragraph or two in the paper discussing such things.
Author Response
We thank the reviewer for his/her valuable comments that have helped to improve the quality and clarity of the manuscript. We have tried our best to address all the comments point-to-point. Please see the details in the attached file.

Reviewer 2 Report
This paper provide the seasonal variations of gravity wave momentum flux at Mohe, China, a mid-high latitude location. After spending a lot of content to test the validity of their data in estimation of GW momentum flux, they finally show the seasonal and interannual variation of momentum flux. The result is attracting, but the paper needs much improvement before it is suitable for publication
Major comments: the author spend a large amount of content to test and prove the validity. Resultingly, their content on the seasonal and interannual variation of GW momentum flux is much less. However, their title is about the GW momentum flux, not the test and validation. For example, they shall also compare with their results with previous studies. There is no 10-year statistic of GW momentum flux before?? I do not think so. I suggest the author expand their GW momentum flux result by comparing with both local ground and satellite observations of GW momentum flux.
For the explanation, they also briefly mention the GW filtering theory, which is far less sufficient to explain the results. Note that Mohe is a mid-high latitude location, maybe the geomagnetic activity also modulates the seasonal variation. Also the solar cycle may contribute their roles. Additionally, the background tide and planetary waves may also contribute. The author can expand the explanation by these aspects. At least we shall know the background waves seasonal and annual variations corresponding to the GW momentum flux
Line 24 meteor radar at Mohe
Line 41 replace ‘depose’ with ‘deposit’
In the introduction, the author shall also acknowledge another major method of calculating the GW flux: the temperature perturbation method proposed by Erns, 2004. The process can be seen in the Appendix of Ern, 2004. Note that this is always suitable for the middle-frequency GWs. The reference Mindy et al. 2020 in the introduction is using this method.
Ern, M., Preusse, P., Alexander, M. J., and Warner, C. D. (2004), Absolute values of gravity wave momentum flux derived from satellite data, J. Geophys. Res., 109, D20103, doi:10.1029/2004JD004752.
The author shall also cite some previous papers using the temperature perturbation method based on local ground observations. Although single local ground observation cannot provide the ratio of wavenumber, it can still indirectly and partly provide the behavior of GW momentum flux
Baumgarten, K., Gerding, M., Baumgarten, G., and Lübken, F.-J.: Temporal variability of tidal and gravity waves during a record long 10-day continuous lidar sounding, Atmos. Chem. Phys., 18, 371–384, https://doi.org/10.5194/acp-18-371-2018, 2018.
Cai, X., Yuan, T., & Liu, H.-L. (2017). Large Scale Gravity Waves perturbations in mesosphere region above northern hemisphere mid-latitude during Autumn-equinox: A joint study by Na Lidar and Whole Atmosphere Community Climate Model. Annales Geophysicae, 35, 181–188. https://doi.org/10.5194/angeo-35-181-2017
Lines 101-103 The data with smaller zenith angle would induce larger error in the estimation of horizontal wind. Larger zenith angle can lead to bigger uncertainty in height measurements.
Line 106 counts reach maximum at dawn
In the data and methodology part, the author shall state that whether the radar can work diurnally or just work during nighttime
The description of Fig1b says the local time variation of XX, but why the x axis is the universal time??
Line 164-170 the author shall provide the meaning of all characters in the equation. Now not all of the characters are given meanings, such as the one in the bracket in sin, sin(2*pi(t-??))
Line 172-176 how do you set the amplitude? Based on what dispersion relationship?? Or you just set them randomly?? How do the amplitude change with altitude? I suggest the author put the figure of GW and the background in a Figure, not just give description, which is not clear.
Line 229 which should arise
Line 238 have obvious variations
Line 245 the magnitude in 2012
Line 261 Analysis of the results by Lomb-scargle is shown in Figure 10
Author Response
Point 1: the author spend a large amount of content to test and prove the validity. Resultingly, their content on the seasonal and interannual variation of GW momentum flux is much less. However, their title is about the GW momentum flux, not the test and validation. For example, they shall also compare with their results with previous studies. There is no 10-year statistic of GW momentum flux before?? I do not think so. I suggest the author expand their GW momentum flux result by comparing with both local ground and satellite observations of GW momentum flux.
Response 1: Thanks for the comment. To our knowledge, some meteor-radar studies analyzed continuous GW momentum fluxes over 10 years, but not to much. Pramitha et al. (2019) use meteor radars in India to study the GW momentum fluxes, and the Thumba station provides continuous estimations over ten years. At the present stage the Mohe meteor radar has one of the best in China that continuously provide high-quality data during the last decade, and we believe that the radar station achieves the top level worldwide. The operation is the results of great and painstaking efforts from our engineers and scientists.
Comparison with other work or model result is truly important. Apologize that we might not take much attention in the original manuscript. According to the reviewer’s comment, we add an additional “Discussion” part before the conclusion to address the comparison and scientific discussion on the variability of GW momentum fluxes.
Point 2: For the explanation, they also briefly mention the GW filtering theory, which is far less sufficient to explain the results. Note that Mohe is a mid-high latitude location, maybe the geomagnetic activity also modulates the seasonal variation. Also the solar cycle may contribute their roles. Additionally, the background tide and planetary waves may also contribute. The author can expand the explanation by these aspects. At least we shall know the background waves seasonal and annual variations corresponding to the GW momentum flux
Response 2: Appreciate the comment that could elevate significance of this paper. We added one figure to illustrate the the background winds (Figure 12), and added one "Discussion" part to discuss the potential contributor as the reviewer's suggestion.
Point 3: In the introduction, the author shall also acknowledge another major method of calculating the GW flux: the temperature perturbation method proposed by Erns, 2004. The process can be seen in the Appendix of Ern, 2004. Note that this is always suitable for the middle-frequency GWs. The reference Mindy et al. 2020 in the introduction is using this method.
Response 3: Thanks for the reference recommendation. We revised the Introduction and include the reference of Ern et al. (2004) as following:
"Satellite has advantage in the spatial coverage. For example, Ern et al. (2004), for the first time, provided the global map of GW momentum fluxes based on CRISTA satellite observations of temperature, according to the polarization relations."
Point 4: The author shall also cite some previous papers using the temperature perturbation method based on local ground observations. Although single local ground observation cannot provide the ratio of wavenumber, it can still indirectly and partly provide the behavior of GW momentum flux.
Response 4: Thanks for the reference recommendation which can enrich the introduction part. We cited the two references as the reviewer's suggestion and introduced them as following:
“Ground-based temperature observation also enable to detect the temperature perturbations (Baumgarten et al., 2018; Cai et al., 2017) and have the potential to derive the GW momentum fluxes using the Ern et al. (2004) method. ”
Point 5: In the data and methodology part, the author shall state that whether the radar can work diurnally or just work during nighttime
Response 5: Thanks for the suggestion. We added one sentence to explain the reviewer's concern as following:
"The meteor radar enable operate for both daytime and nighttime continuously and generally undisrupted by the severe weather condition."
Point 6: The description of Fig1b says the local time variation of XX, but why the x axis is the universal time??
Response 6: Thanks for the comment that make our presentation much clearer. We are sorry for the mistake that the Figure 1b and its description is not matched. We revised the x-axis to be local time.
Point 7: Line 164-170 the author shall provide the meaning of all characters in the equation. Now not all of the characters are given meanings, such as the one in the bracket in sin, sin(2*pi(t-??))
Response 7: Thanks for the comment that we do not present clear enough. We have added the description on the symbols which we do not given the meanings.
”(ki, li, mi) are the zonal, meridional, and vertical wavenumber of the i-kind GW, and TGWi describes the corresponding GW period. The symbols of x, y, and z indicate the location of detected meters in east, north, and vertical direction taken the radar as the coordinate origin. θ and φ are the zenith and azimuth angle. t is the universal time and (δUD, δUSD, δU2D; δVD, δVSD, δV2D) indicate the phase of the diurnal tides, semidiurnal tides, and 2-day PWs in zonal and meridional winds, respectively. (TD, TSD, T2D) are the periods of diurnal tides, semidiurnal tides, and 2-day PWs. “
Point 8: Line 172-176 how do you set the amplitude? Based on what dispersion relationship?? Or you just set them randomly?? How do the amplitude change with altitude? I suggest the author put the figure of GW and the background in a Figure, not just give description, which is not clear.
Response 8: Thanks for the comment. We apologize that specification on these parameters are not much clear. The amplitudes are set to be typical value and dispersion relationship is not considered here. In Case 1, 2, and 4, the amplitude is set to be not changed with altitude. According to the reviewer's suggestion, we added one figure (Figure 3 in revised manuscript) to illustrate the prescribed GWs and the background winds. Additional description is also include in the manuscript as following:
"Here, an example of prescribed wind fields with and without GWs is shown in Figure 3. The example is corresponding to Case 2 and the angular information of detected meteors is for the month of June 2012. Obvious wave structures could be seen in Figures 3a and 3c, and our aim is to extract the information of GW momentum fluxes from such patterns."
Point 9: other text corrections.
Response 9: Thanks for the careful examination on the typos, grammar errors, and misunderstanding words throughout the draft. We have checked and corrected them according to the reviewer's suggestions.
Reviewer 3 Report
The paper entitled “Decadal continuous meteor-radar estimation of mesopause gravity wave momentum fluxes over Mohe (53.5°N, 122.3°E): capability evaluation and interannual variation” by Zhou et al., calculated gravity wave momentum fluxes using decadal continuous observations by Mohe (53.5°N, 122.3°E) meteor radar. They evaluated the capability of meteor radar in retrieving gravity wave momentum fluxes using Hocking’s method in the MCD way. Each case of simulated and retrieved gravity wave momentum fluxes are matching well. The paper is organized well and I recommend a minor revision.
Comments:
-
Line 100… Generally, the zenith angle values take from the range of 15-45 degrees.
-
Line 243…. What may be the possible reason for the strong zonal momentum fluxes in 2012?
-
Please elaborate on the discussion section.
Author Response
Point 1: Line 100… Generally, the zenith angle values take from the range of 15-45 degrees.
Response 1: Thanks for the comment. It is true that some authors took the zenith angle within 15–45° (e.g. Hocking, 2005; Pramitha et al., 2019). We also found that some other authors took 50° as the upper limit (e.g. de Wit et al., 2015, 2016; Liu et al., 2013; Placke et al., 2011). Vincent et al. (2010) even only discarded the zenith angle larger than 80°. We chose the zenith angle within 15–60° here mainly because some previous work used that (e.g. Jia et al., 2018; Tian et al., 2020, 2021). To clarify this point, we added a sentence in Line 103-104:
Line 103-104: The selection criterion of zenith angle is slightly different in previous works, and here we follow the works by Jia et al. (2018) and Tian et al. (2020, 2021)..
Point 2: Line 243…. What may be the possible reason for the strong zonal momentum fluxes in 2012?
Response 2: Thanks for the comment. According to the reviewer's comment, we examined the wind profile below 80 km from the ERA-5 reanalysis data. The mesospheric summer westward jet (~60-70 km) shows that no significant enhancement in 2012. So the enhanced GW momentum fluxes should be came from the enhanced wave sources, but unfortunately we do not have solid observational evidence to support this argument at the present stage. We added some discussion about it in the manuscript.
Point 3: Please elaborate on the discussion section.
Response 3: Thanks for the suggestion. We added an discussion part to discuss some scientific arguments, mainly including the middle atmospheric dynamics and comparison with other works.
Reviewer 4 Report
The paper entitled "Decadal continuous meteor-radar estimation of mesopause gravity wave momentum fluxes over Mohe (53.5°N, 122.3°E): capability evaluation and interannual variation" shows evaluation of capability of Mohe meteor radar in retrieving gravity wave (GW) momentum fluxes and discusses the retrieved GW variability for middle and upper atmospheric physics. Estimating GW momentum fluxes accurately is significant study for better understanding known/unknown behavior of the global/planetary atmosphere.
The authors tried to use the decadal datasets obtained over Mohe, rather higher latitude place in middle latitude region, for investigating GW momentum fluxes for mesopause altitude region and summarized no obvious Quasi-Biennial Oscillation (QBO) like signal was found there, even when the method applied here is appropriate.
The study shows 4 case retrieval calculations with showing precise parametric definition shown in Table 1, and clear results are shown in Figures 3-6 for July, 2012 case, as well as summarized in Figure 7 for decadal calculations with standard deviation so as to assess the capability of this kind of method for mesopause region.
Conclusion of the capability of applied retrieval method looks good in the altitude range between roughly 84 and 94 km with clear standard deviation distribution shown in Figure 7.
Thus, the submitted version article is considered almost acceptable. Please read the comments and correct some minor descriptions in the text and attached Table and Figures.
Minor comments/questions:
The reviewer thought that Figure 7 shows the most significant summarized information in the results obtained in this study. There are two curves for each panel for standard deviations, and only for Case 1, the red curve indicates larger than blue one. It looks very interesting. If the authors can explain why it is, please add some discussion about this issue in the text.
Please identify all the symbols (variables) shown in Table 1 and Equations used. There are no explanations for R0, R1, and R2 in Table 1. The reviewer cannot understand the meanings of 21 (20) just before R1 (R2) in the footnote explanations for functions F4(t) and G4(t).
In Figure 8c and 8d, there are red regions for higher GW momentum fluxes coming from lower atmosphere in summer time. It seems the observatory condition of 53.5 degrees North becomes near the sub-solar point of the planet at summer solstice. Thus, the delay of the maximum of the red region than the summer solstice of late June seems interesting parameter for transporting GW momentum fluxes. If the authors understand any available references for explaining this issue, please add some comments on the text.
The reviewer recently searched on internet about the paper(s) concerning Mohe observatory meteor radar datasets and found the paper Gong et al. (2022) recently published as JGR Space Physics journal, but it was not referred for this submission. The paper shows interseasonal oscillations (ISO) in the MLT region during 2015/2016 winter with showing relationship between tropospheric MJO and ISO over Mohe. The authors of both article are totally independent between each other, so the reviewer does not know the relationship between them. On Gong et al. (2022), they referred Yu et al. (2013) for detailed description about Mohe meteor radar, and the same paper is referred on this submission. Thus, if the authors do not mind adding one more reference, please consider it for showing good relationship between the both scientific teams.
Line 229: the different angular location of the detected meteors.
The reviewer felt the description might be a bit vague for clear understanding by readers, rather than the other clear descriptions in whole of the text.
Tips for corrections:
Abstract:
Line 34: QBO-like -> Quasi-Biennial Oscillation (QBO) like
or QBO (Quasi-Biennial Oscillation) -like
Main text:
Line 88: Momentum -> momentum
Line 101: 200m/s -> 200 m/s
Line 103: measurement -> determination (?)
Line 106: 18UT -> 22UT (if LT=UT+8 hr at Mohe)
For equations (1)-(4): please define all the shown variables, especially for the undefined parameter of k, l, m, T, δ, θ, φ, USD, and VSD.
Line 185: give -> gives
Line 201: tides The -> tides. The
Line 216: bad data -> It sounds a bit rough description.
Line 229: should be arise -> should be arisen
Line 234: assess. -> assessed.
Line 242: in both zonal and momentum fluxes. -> in both zonal and meridional momentum fluxes.
Line 255: become -> becomes
Line 261: further Analyze -> further analyze
Line 273: show -> shows
Table 1:
"TGW1, TGW2," for Case 3: 20, 30 -> 20, 30,
Footnote: each days -> each day
Figure 1:
Please add two red small arrows from bottom of panel to the lowest floor of appeared curves for 10UT and 22UT.
Figure 3:
Caption: respectively. (b). -> respectively (b).
Figure 5:
Caption: Case 3 -> Case 3.
Figure 6:
Caption: case 4. -> Case 4.
Figure 7:
Caption: cases 1-4. -> Cases 1-4.
Figure 10:
The reviewer recommend the authors to reduce the shown horizontal information to be within 120 months because they used 10-year (120-month) length datasets only.
Author Response
We thank the reviewer for his/her valuable comments that have helped to improve the quality and clarity of the manuscript. We have tried our best to address all the comments point-to-point as detailed in the following.
Point 1: The reviewer thought that Figure 7 shows the most significant summarized information in the results obtained in this study. There are two curves for each panel for standard deviations, and only for Case 1, the red curve indicates larger than blue one. It looks very interesting. If the authors can explain why it is, please add some discussion about this issue in the text.
Response 1: Thanks for the comment. It is exactly an interesting point that we do not recognized before. Add some discussion would be helpful to improve the paper quality. However, to our knowledge, we are sorry for that there is not a plausible explanation at the present stage. So, we decided to remind the reader about it in the main text as follow.
Line 234-236: The estimation uncertainty of meridional momentum fluxes is slightly larger than that of zonal momentum fluxes only except for Case 1.
Point 2: Please identify all the symbols (variables) shown in Table 1 and Equations used. There are no explanations for R0, R1, and R2 in Table 1. The reviewer cannot understand the meanings of 21 (20) just before R1 (R2) in the footnote explanations for functions F4(t) and G4(t).
Response 2: Thanks for the comment pointing out where we need to clarify. We have add the explanation on the meaning of the symbols in the main text as the reviewer's suggestion.
Line 216-219: As shown in Table 1, the introduced intermittent GW1 and GW2 are described by random functions of F4 and G4, which means the GW1 and GW2 would occur randomly in one day (24 hours) and lasting 3 and 4 hours, respectively. In the rest of 20 and 21 hours, the GW1 and GW2 do not exist.
Point 3: In Figure 8c and 8d, there are red regions for higher GW momentum fluxes coming from lower atmosphere in summer time. It seems the observatory condition of 53.5 degrees North becomes near the sub-solar point of the planet at summer solstice. Thus, the delay of the maximum of the red region than the summer solstice of late June seems interesting parameter for transporting GW momentum fluxes. If the authors understand any available references for explaining this issue, please add some comments on the text.
Response 3: Thanks for the comment. It is interesting phenomenon that the maximum of GW momentum fluxes is generally occurred in July rather during exactly the June solstice. We have examined the ERA-5 reanalysis data that covers the wind profile below 80 km. The result shows that the maximum of westward wind in the mesosphere is also generally occurred in July rather than June. It is consistent with the annual variation of the zonal GW momentum fluxes. As the mesospheric circulation is determined by not only the radiative effect but also the dynamical processes (e.g. eddy forcing). Thus, we decided to add some context to discuss this point in the manuscript.
Point 4: The reviewer recently searched on internet about the paper(s) concerning Mohe observatory meteor radar datasets and found the paper Gong et al. (2022) recently published as JGR Space Physics journal, but it was not referred for this submission. The paper shows interseasonal oscillations (ISO) in the MLT region during 2015/2016 winter with showing relationship between tropospheric MJO and ISO over Mohe. The authors of both article are totally independent between each other, so the reviewer does not know the relationship between them. On Gong et al. (2022), they referred Yu et al. (2013) for detailed description about Mohe meteor radar, and the same paper is referred on this submission. Thus, if the authors do not mind adding one more reference, please consider it for showing good relationship between the both scientific teams.
Response 4: Thanks for the suggestion about additional information enriching this paper. We have added the reference of Gong et al. (2022) and add some sentence to discuss the relationship.
Point 5: Line 229: the different angular location of the detected meteors.
The reviewer felt the description might be a bit vague for clear understanding by readers, rather than the other clear descriptions in whole of the text.
Response 5: Thanks for the suggestion. We have revised the phrase to be "angular information of the detected meteors", which matches the context.
Point 6: Tips for corrections on text
Response 6: Thanks for the careful examination on the typos, grammar errors, and misunderstanding words throughout the draft. We have checked and corrected them according to the reviewer's suggestions. The x-axis limitation of Figure 10 is also redrawn.
Reviewer 5 Report
remarks
97 word radar is used twice. Is it correct?
128, 130, 145 PW -it is not code
176 amplitude -what units
Author Response
We thank the reviewer's suggestions about this paper. The point-by-point response is addressed as below.
Point 1: 97 word radar is used twice. Is it correct?
Response 1: Thanks for point out the typo. We have deleted the second word of "radar".
Point 2: 128, 130, 145 PW -it is not code
Response 2: Thanks for the comment. We have added the full name of "PWs"->"planetary waves" at where it appears at the first time (Line 128)
Point 3: 176 amplitude -what units
Response 3: Thanks for the suggestion. We have add the units here, which is "m/s" .
Round 2
Reviewer 1 Report
2nd Review on:
Decadal continuous meteor-radar estimation of mesopause gravity wave momentum fluxes over Mohe (53.5°N, 122.3°E): capability evaluation and interannual variation
By
Xu Zhou, Xinan Yue, Libo Liu, You Yu, Feng Ding, Zhipeng Ren, Yuyan Jin and Hanlin Yin
General Comment:
The manuscript compares two methods to obtain momentum fluxes by applying the algorithm proposed by Hocking et al., 2005. The method is compared to synthetic data, which is almost identical to the cases discussed in Fritts et al., 2010. In summary, the paper presents results based on 10 years of data collected with the Mohe meteor radar. The revised manuscript added some discussion of the results with a strong focus on Asian publications. The reviewer acknowledges this improvement, however, it is still recommended to add the other Reynolds stress terms and to discuss these results as well. Furthermore, the reviewer suggests a careful assessment about the usage of the term retrieval (mathematical).
General Comment:
Retrievals are a dedicated mathematical approach that is often applied to satellite observations and irradiance measurements. The reviewer is not aware that such techniques have been applied in any of the cited literature throughout the manuscript. There the momentum fluxes are always obtained by applying least-square fits. The mathematical approach of retrievals goes back to Shannon and Waever 1948/49 and is based on information theory and the maximum entropy concept. For more details please read:
Shannon, C. E.: A Mathematical Theory of Communication, Bell Syst. Tech. J., 27, 379–423, https://doi.org/10.1002/j.1538-7305.1948.tb01338.x, 1948.
Shannon, C. E. and Weaver, W.: The Mathematical Theory of Communication, University of Illinois Press, Urbana, IL, 1949.
Rodgers, C. D.: Inverse Methods for Atmospheric Sounding: Theory and Practice, World Scientific, Singapore, ISBN 981-02-2740-X, 2000.
The reviewer suggests to not use the term-retrieval when talking about the data analysis of the momentum fluxes, although many publications claim to use a retrieval, the more appropriate term is least-square fit.
Detailed comments:
Line 61: Please add Hocking, 2005 to the references for meteor radar. Fritts et al., 2010 applied the method, but the idea and first results go back to the original study.
Line 66: Please update the Baumgarten et al., 2018 reference to the most recent version (https://angeo.copernicus.org/articles/37/581/2019/).
Lines 70-89: This paragraph should contain a statement about the two meteor concepts. The SAAMER and Trondheim meteor radars were developed to ensure more reliable MF measurements compared to the standard meteor radars. In fact, the MF-meteor radars have multiple beams focusing more energy close to the zenith increasing the reliability of the measurements. In particular, the ability to observe robust and resilient vertical velocity fluctuations is key and one of the weak points when applying Hocking’s method to standard meteor radars.
The second important aspect is that Hocking’s method solves for the complete Reynold stress tensor entries and also presents <u’v’> and u’2, v’2, w’2. It is not possible to derive only the other two fluxes. Thus, it is recommended to show these other parameters and their seasonal variability as well. Maybe they can show histograms of w’2 and explain/discuss why there are negative values of w’2.
Line 200-204:
This paragraph is problematic. The paper assumes a GW with specified properties for a given time t, but plots the information using all detections from one month. This is somehow contradicting. The Figure makes no sense in this context. It is recommended to redo the figure and only uses the meteors for an interval t+/- dt with dt<5 min. Or it is possible to perform a simulation using all data within 2 hours and include the GW propagation assuming a certain phase velocity.
Author Response
We thank the reviewer for his/her valuable comments that have helped to improve the quality and clarity of the manuscript. We have tried our best to address all the comments point-to-point as detailed in the following.
Point 1: The reviewer suggests to not use the term-retrieval when talking about the data analysis of the momentum fluxes, although many publications claim to use a retrieval, the more appropriate term is least-square fit.
Response 1: Thanks for the suggestion. We have changed the words of "retrieval/retrieved/retrieving" throughout the manuscript.
Point 2: Please add Hocking, 2005 to the references for meteor radar. Fritts et al., 2010 applied the method, but the idea and first results go back to the original study.
Response 2: Thanks for the comment. We have added the Hocking's reference here.
Point 3: Please update the Baumgarten et al., 2018 reference to the most recent version (https://angeo.copernicus.org/articles/37/581/2019/).
Response 3: Thanks. We have updated the reference.
Point 4: This paragraph should contain a statement about the two meteor concepts. The SAAMER and Trondheim meteor radars were developed to ensure more reliable MF measurements compared to the standard meteor radars. In fact, the MF-meteor radars have multiple beams focusing more energy close to the zenith increasing the reliability of the measurements. In particular, the ability to observe robust and resilient vertical velocity fluctuations is key and one of the weak points when applying Hocking’s method to standard meteor radars. The second important aspect is that Hocking’s method solves for the complete Reynold stress tensor entries and also presents <u’v’> and u’2, v’2, w’2. It is not possible to derive only the other two fluxes. Thus, it is recommended to show these other parameters and their seasonal variability as well. Maybe they can show histograms of w’2 and explain/discuss why there are negative values of w’2.
Response 4: Thanks for the valuable comment. For the first aspect, we have added
Point 5: This paragraph is problematic. The paper assumes a GW with specified properties for a given time t, but plots the information using all detections from one month. This is somehow contradicting. The Figure makes no sense in this context. It is recommended to redo the figure and only uses the meteors for an interval t+/- dt with dt<5 min. Or it is possible to perform a simulation using all data within 2 hours and include the GW propagation assuming a certain phase velocity.
Response 5: Thanks.
Reviewer 2 Report
The author has done great jobs in improving the paper. The response to reviewer has answered all my questions and concerns. The discussion added is very sufficent and provide a very good analysis and comparison of their results. I recommend the paper to have a minor revision before it is suitable for publication
Line 62 Satellite has advantages
Line 65 Ground-based Lidar observations also enable to XX
Line 97 Our work will give a comprehensive analysis
Line 119-120 The meteor radar enables operation for both daytime and nighttime
For figure 9, I suggest the author put a sub-plot of Kp and F10.7 during these years to show solar and geomagnetic variations for reference. Since the author mentioned the GW momentum flux variations with the increase of solar activity
Author Response
Appreciate for the reviewer's help. Here are point-to-point reply for the comments and suggestions
Point 1: Text corrections.
Response 1: We have corrected them as the reviewer's suggestions.
Point 2: For figure 9, I suggest the author put a sub-plot of Kp and F10.7 during these years to show solar and geomagnetic variations for reference. Since the author mentioned the GW momentum flux variations with the increase of solar activity
Response 2: Thanks for the comment. We have added a penal in Figure 9 that shows the monthly variations of F107 and Ap index.